# Repeats in S1 Proteins: Flexibility and Tendency for Intrinsic Disorder

**DOI:** 10.3390/ijms20102377

**Published:** 2019-05-14

**Authors:** Andrey Machulin, Evgenia Deryusheva, Mikhail Lobanov, Oxana Galzitskaya

**Affiliations:** 1Skryabin Institute of Biochemistry and Physiology of Microorganisms, Russian Academy of Sciences, Federal Research Center “Pushchino Scientific Center for Biological Research of the Russian Academy of Sciences, 142290 Pushchino, Russia; and.machul@gmail.com; 2Institute for Biological Instrumentation, Federal Research Center “Pushchino Scientific Center for Biological Research of the Russian Academy of Sciences, 142290 Pushchino, Russia; evgenia.deryusheva@gmail.com; 3Institute of Protein Research, Russian Academy of Sciences, 142290 Pushchino, Russia; mlobanov@phys.protres.ru

**Keywords:** ribosomal proteins S1, structural domains, intrinsically flexibility, FoldUnfold program, IsUnstruct program

## Abstract

An important feature of ribosomal S1 proteins is multiple copies of structural domains in bacteria, the number of which changes in a strictly limited range from one to six. For S1 proteins, little is known about the contribution of flexible regions to protein domain function. We exhaustively studied a tendency for intrinsic disorder and flexibility within and between structural domains for all available UniProt S1 sequences. Using charge–hydrophobicity plot cumulative distribution function (CH-CDF) analysis we classified 53% of S1 proteins as ordered proteins; the remaining proteins were related to molten globule state. S1 proteins are characterized by an equal ratio of regions connecting the secondary structure within and between structural domains, which indicates a similar organization of separate S1 domains and multi-domain S1 proteins. According to the FoldUnfold and IsUnstruct programs, in the multi-domain proteins, relatively short flexible or disordered regions are predominant. The lowest percentage of flexibility is in the central parts of multi-domain proteins. Our results suggest that the ratio of flexibility in the separate domains is related to their roles in the activity and functionality of S1: a more stable and compact central part in the multi-domain proteins is vital for RNA interaction, terminals domains are important for other functions.

## 1. Introduction

It is known that multi-domain proteins are frequently characterized by the occurrence of domain repeats in proteomes across the three domains of life: Bacteria, Archaea, and Eukaryotes [1,2]. Proteins with repeats participate in nearly every cellular process from transcriptional regulation in the nucleus to cell adhesion at the plasma membrane [3]. In addition, due to their flexibility, domain repeats can be found in cytoskeleton proteins, proteins responsible for transport and cell cycle control [4]. Proteins with structural repeats are believed to be ancient folds.

One such unique protein family is a family of bacterial ribosomal proteins S1 in which structural domain S1 (one of the oligonucleotide/oligosaccharide-binding fold (OB-fold) options) repeats and changes in a strictly limited range from one to six [5]. As demonstrated in our recent paper [5], the family of polyfunctional ribosomal proteins S1 contains about 20% of all bacterial proteins, including the S1 domain. This fold also could be found in different eukaryotic protein families and protein complexes in different number variations. Such multiple copies of the structure increase the affinity and/or specificity of the protein binding to nucleic acid molecules.

Recently we have shown that the sequence alignments of S1 proteins between separate domains in each group reveal a rather low percentage of identity. In addition, the verification of the equivalence of the domain characteristics showed that for long S1 proteins (five- and six-domain containing S1 proteins) the central part of the proteins (the third domain) is more conservative than the terminal domains and apparently is vital for the activity and functionality of S1. Data obtained indicated that for general functioning of these proteins, the structure scaffold (OB-fold) is obviously more important than the amino acid sequence [6]. This statement is in good agreement with the fact that there is a high degree of conservatism and topology position of the binding site on the OB-fold surface in others proteins, as well as “fold resistance” to mutations and the ability to adapt to a wide range of ligands, which allows us to consider this fold as one of the ancient protein folds. For example, the author of article [7] proposed considering this core structure of inorganic pyrophosphatase as the evolutionary precursor of all other superfamilies.

At present, the structure of S1 from *Escherichia coli* was obtained only with a very low resolution of 11.5 Å using cryo-electron microscopy [8]. In the Protein Data Bank, there are only 3D structures of separate domains of ribosomal S1 from *E. coli* obtained by NMR [9,10]. Recently, protein S1 on the 70S ribosome was visualized by ensemble cryo-electron microscopy [11]. It was shown that S1 cooperates with other ribosomal proteins (S2, S3, S6, and S18) to form a dynamic mesh near the mRNA exit and entrance channels to modulate the binding, folding and movement of mRNA. The cryo-electron microscopy was also used to obtain the structure of the inactive conformation of the S1 protein as part of a hibernating 100S ribosome [12].

A separate S1 domain from the ribosomal proteins S1 [9] and other bacterial proteins containing an S1 domain [13,14,15,16] represents a β-barrel with an additional α-helix between the third and fourth β-sheets. As shown in the articles [13,14,15,16], the S1 domain as a part of different bacterial proteins (as well as in eukaryotic proteins) itself is quite compact, therefore it crystallizes and is visualized very well.

At the same time, there are currently no determined structures for full-length, intact ribosomal S1 proteins containing a different number of structural domains (six in *E. coli*, five in *Thermus thermophilus*, etc.). This may be due to the increased flexibility of multi-domain proteins as was noted in [17]. In addition, some biochemical studies suggest that in solution and on the ribosome, S1 can have an elongated shape stretching over 200 Å long [17,18,19,20].

Moreover, recently it was shown that the prediction of intrinsic disorder within proteins with the tandem repeats supports the conclusion that the level of repetition correlates with their tendency to be unstructured and the chance to find natural structured proteins in the Protein Data Bank (PDB) increases with a decrease in the level of repeat perfection. Also, the authors suggested that in general, the repeat perfection is a sign of recent evolutionary events rather than of exceptional structural and/or functional importance of the repeat residues [21].

Despite all these observations, the flexibility of S1 proteins, their tendency for intrinsic disorder, and the structural characteristics of this family have not been studied as of yet. To fill this gap, we have analyzed here the flexibility of the bacterial S1 proteins within and between structural domains, as well as the tendency for intrinsic disorder of the S1 protein family.

## 2. Results and Discussion

### 2.1. Analysis of Tendency for Intrinsic Disorder of the Bacterial S1 Proteins

Binary disorder analysis using the charge–hydrophobicity plot cumulative distribution function (CH-CDF) plot [22] showed that most of the bacterial S1 proteins (1374 sequences) (53%) are expected to be mostly ordered (or folded, ‘F’) (Figure 1a).

Mixed or molten globular (‘MG’) forms comprised the remaining 47% of the bacterial S1 proteins. Major protein states for separate groups of the S1 proteins (different number of structural domains) according to the CH-CDF analysis are shown in Figure 1b. In the case of S1 proteins containing one, two or six structural domains (1S1, 2S1, 6S1) the ordered state prevailed (83%, 78% and 67%, respectively). S1 proteins containing three, four and five domains were classified as molten globule state according to the CH-CDF analysis in 69%, 74% and 56% cases, respectively. It was seen that with an increase in the number of structural domains (starting from the three-domain containing proteins), the MG state prevailed, but for six-domain proteins only 34% of the records belonged to this area. Despite the fact that one-domain and two-domain containing proteins were the least represented in our dataset, the data obtained for these groups results are in good agreement with the fact that the separate S1 domain is stable and has rather rigid structure [13,14,15,16]. Note that for other structural variants of the OB-fold (for example, CSD domain [23], inorganic pyrophosphatase [24], *MOP-like* [25], etc.) there are available structures that also have only one or two (repeated) domains [5].

### 2.2. Analysis of Intrinsic Flexibility and Disorder of the Bacterial S1 Proteins and Its Domains.

For analysis of intrinsic flexibility and disorder of the full length bacterial S1 proteins and its separate structural domains we used the FoldUnfold (average window 11 aa and 5 aa) and IsUnstruct programs; their possibilities and accuracy were described in [26,27,28,29]. The obtained results are given in Table 1.

Analysis of the percentage of disorder in the full length S1 proteins and in their separate domains by the FoldUnfold (average window 11 aa and 5 aa) and IsUnstruct programs revealed their close similarity (Table 1).

For full-length proteins, the highest percentage of disorder was detected for four- (30%) and five-domain (30%) containing proteins using the FoldUnfold program (average window 5 aa). The smallest percentage was in the six-domain proteins (13%) when using the FoldUnfold program (average window 11 aa). This indicates the predominance of relatively short flexible or unstructured regions in the considered sequences of the proteins of this group, consistent with the fact that the binary predictor of the CH-CDF plot revealed the ordered states for 67% of proteins in this group.

Most of the separate S1 domains exhibited disorder values around 20%. The lowest percentage of disorder (except the third domain in three-domain containing proteins and the separate domains in the one-domain containing proteins) predicted by the FoldUnfold program (average window 5 aa) was the third domain in six-domain containing proteins (13%). Using the FoldUnfold program (average window 11 aa) and IsUnstruct for this domain also revealed a relatively low percentage of intrinsically disorder compared with other domains in this group and other groups (by the number of domains), 19% and 21%, respectively. The largest percentage of disorder predicted by the IsUnstruct program belonged to the sixth domain in the six-domain containing proteins (45%). Using the FoldUnfold program for six-domain containing proteins, a propensity for a more disordered state in the terminal domains was also identified. Note that, earlier, we have shown that for long S1 proteins (six-domain S1 proteins) the central part of the proteins (the third domain) is more conservative (as a percent of identity between separate domains) than the terminal domains, and apparently is vital for the activity and functionality of S1 proteins [6].

The concept of order and disorder in protein segments has often been investigated in correlation with the presence or absence of protein repeats at the sequence level. It is noticed that intrinsically disordered proteins often correspond to regions of low compositional complexity (low sequence entropy) and sometimes to repetitive sub-sequences, for example, in fibrillar proteins [30]. Also in some special cases, protein repeats (for example, in the PEVK ((Pro-Glu-Val-Lys) domain) regions of human titin, the prion proteins, or the CTD domain of RNA polymerase) are discussed in detail [31]. However, these findings on specific instances are hard to generalize. A general property observed is that a higher level of repeat perfection correlates positively with the disordered state of protein sub-chains [21].

S1 proteins, having a low degree of conservatism (not perfect repeats) [6], in addition to the found low degree of disorder within and between the domains, demonstrate the unique structural organization of proteins of this family. Apparently, the organization is closer to the formation of the quaternary structure of globular proteins, with the same structural organization of individual structural domains.

### 2.3. Flexibility of S1 Domain in the Bacterial Proteins

Besides the ribosomal proteins, S1 domains are identified in different quantities in different archaeal, bacterial and eukaryotic proteins [5]. As we recently showed, archaeal proteins contain one copy of the S1 domain, while the number of repeats in the eukaryotic proteins varies between 1 and 15 and correlates with the protein size. In the bacterial proteins, the number of repeats is no more than 6, regardless of the protein size. To compare the obtained data on the flexibility of ribosomal proteins S1, S1 domains from some bacterial proteins [5] were investigated using the approaches described above (Table 2).

In all proteins (Table 2, Figure 2), one S1 domain was identified and had a low degree of disorder (about 20%). It can be seen that when the size of average window of the FoldUnfold program decreases, this percentage increases, indicating the presence of flexible sections of short length in the considered proteins. This is consistent with the fact that S1 domains in these proteins are well determined by various methods (Figure 2).

However, structures of proteins containing three or more S1 domains have not been determined yet. In the eukaryotic proteins containing more than two S1 domain (from 7 to 15) determined structures also are not available. Note that in these proteins, functions of separate S1 domains are not defined, for example, Rrp5p [32], Prp22p [33].

### 2.4. Analysis of the Ratio of Secondary Structures in the Bacterial S1 Proteins and Its Domains

Obtained ratios of regions connecting secondary structure according to the JPred predictions are shown in Table 3.

It can be seen that the ratio of regions connecting the secondary structure in separate domains was approximately the same and equal to about 50%, which in addition to conservative secondary structure indicates about the same organization of separate S1 domains. For full length proteins this ratio (linkers and regions connecting secondary structures within domains) was also about 50%, indicating about the same organization of multi-domains containing S1 proteins. The average percent of linkers between structural domains was about 30–40%. The obtained results are in a good agreement with the predictions of the FoldUnfold and IsUnstruct programs and CH-CDF plots, and characterized the family of S1 proteins as proteins with relatively short flexible regions within domains and between them that apparently prefer to be in the folded or MG state. In addition to the aforementioned lower conservatism between separate domains in each group, it can be argued that the unique S1 protein family is different in the classical sense from a protein with tandem repeats, such as the ANK family, leucine-rich-repeat proteins, etc. [4]. This family having repeats (separate structural domains) with 70 residues is close to a “beads-on-a-string” organization with each repeat being folded into a globular domain, for example, Zn-finger domains [34], Ig-domains [35] and the human matrix metalloproteinase [36]. Thus, one of the reasons for the absence of allowed three-dimensional structures of multi-domain S1 proteins may be the mobility of domains relative to each other due to the flexibility of interdomain linkers.

In fact, the biochemical experimental study of various fragments allowed establishing the functions of individual protein domains and parts only for the well-studied 30S ribosomal protein S1 with six S1 domain repeats from *E. coli*. For example, it has been shown that cutting one S1 domain from the C-terminus or two S1 domains from the N-terminus of the protein reduces only the effectiveness of protein functions but not its functional abilities; the sixth domain is bound with the process of autoregulation of synthesis, thus cutting off the fifth and sixth domain leads to effective participation of the remaining part of protein only in synthetic mRNA translation [37,38]. Our results indicated about the same organization of separate S1 domains and full-length proteins (conservative secondary structure, ratio of linkers and regions connecting secondary structures within domains). In addition, the percent of intrinsic flexibility is less for the central domains in the multi-domain proteins. These facts allowed us to assume that for all multi-domain S1 proteins more stable and compact domain are located in the central part and are vital for RNA interaction, while more flexible terminals domains are for other functions. The obtained results will be used as a base for investigation of the proposed theories on the evolutionary development of proteins with structural repeats: From the multi-repeat assemblies to single repeat or vice versa.

## 3. Materials and Methods 

### 3.1. Construction of Ribosomal Proteins S1 Dataset

To make a representative dataset of records for the family of ribosomal proteins S1 from the UniProt database, all records for the bacteria containing any one of the keywords «30s ribosomal protein s1», «ribosomal protein s1», «30s ribosomal protein s1 (ec 1.17.1.2)», «30s ribosomal protein s1 (ribosomal protein s1)», «ribosomal protein s1 domain protein», «rna binding protein s1», «rna binding s1 domain protein», «s1 rna binding domain protein» in the protein name were selected (UniProt release 2018_04). Then the obtained array of data was used to choose only proteins encoded by the rpsA gene or its analog; for example, rpsA_1, rpsA_2, rpsA_3, etc. Only this gene, coding the ribosomal protein S1, in the European nucleotide archive (ENA, http://www.ebi.ac.uk/ena) is affiliated to the STD class, that is, the class of standard annotated sequences. From the obtained dataset records, those with six-digital identification numbers (annotated records in the UniProt database) were selected. All data were collected in one file that was the basis for further analysis, namely for collection of data on the number of structural domains and for phylogenetic grouping in the main bacterial phyla (http://bioinfo.protres.ru/other/uniprot_S1.xlsx). Records characterized by the presence of the word “candidate” were removed from our dataset. The automated advanced exhaustive analysis allowed us to choose 1374 records corresponding to these search parameters.

### 3.2. Number and Identification of Structural Domains in Protein Sequences

The values of the number of S1 domains corresponding to the SMART database (about 1200 domains), were selected for each analyzed record. If no data on the number of domains in one of the analyzed bases was available (None), this number was taken to be zero (these records were removed from investigated dataset). Accurate borders for each S1 domain for each record were taken from the UniProt database (position, domain and repeats field).

### 3.3. Prediction of Disordered Regions and Tendency for Intrinsic Disorder

#### 3.3.1. FoldUnfold and IsUnstruct Programs

The FoldUnfold program is accessible at http://bioinfo.protres.ru/ogu/. The principle of its operation is described elsewhere [26,27]. Such a property of residues as the observed average number of contacts in a globular state, closed at a given distance, was used. To predict IDRs (intrinsically disordered regions) in the protein chain using the amino acid sequence, every residue was given an expected number of contacts in the globular state. Then averaging was done by the residue equal to the window width. The obtained average value of expected contacts was ascribed to the central residue in the chosen window. After that the window was shifted by one residue, and the procedure was repeated. On the profile of expected contacts, a boundary was marked that separated structured and unstructured residues. The mean expected number of closed residues, estimated from the sequence, was equal to the sum of expected contact residues divided by the number of amino acid residues in the protein. According to the algorithm of the program, the size of disordered (flexible) regions in such a protein must be equal to or greater than the size of the averaged window. Therefore, the number of predicted regions depended on the window size. The window size in 11 amino acid residues was optimal for the search for relatively short disordered regions in the polypeptide chain. In the case of searching for long disordered regions in partially disordered proteins, the window size must be increased to several tens of amino acid resides. At the same time, for searching for short loops one should use the averaged window size of five amino acid residues, which is optimal for this task.

The IsUnstruct program (v.2.02) is accessible at http://bioinfo.protres.ru/IsUnstruct/. The algorithm of the IsUnstruct program is based on the Ising model. For estimation the energy of any state, the energy of the border between ordered and disordered residues and the energies of initiation of disordered state at the ends were used [39]. After the optimization procedure [28], 20 energetic potentials for residues were obtained which were considered to be in a disordered state, the energy of border, and the energies of initiation of disordered state at the ends. The energy of the completely ordered state was taken to be zero.

#### 3.3.2. CH-CDF Analysis

The charge–hydrophobicity plots (CH-plots) [40] and the cumulative distribution function (CDF) analysis [41] were used for binary prediction of protein stability based of its amino acid sequence.

The Y-coordinate in the CH-CDF plot corresponded to the distance from the obtained ordinate value to the correlation line separating the structured and unstructured conformational state of the protein on the CH (charge-hydrophobicity) plot. The X-coordinate on the CH-CDF plot corresponded to the distance from the obtained ordinate value to the correlation line separating the structured and unstructured conformational state of the protein in the CDF. Thus, in the coordinates of CH-CDF plot it was possible to assign the sequence to one of four quadrants (four conformational states). I quadrant (CH > 0, CDF > 0) were rare proteins for which it was impossible to determine accurately the state (unusual/rare); II quadrant (CH > 0, CDF < 0) were unfolded proteins (U), III quadrant (CH < 0, CDF < 0) was the state of the molten globule (MG), IV quadrant (CH < 0, CDF > 0) were structured proteins (F) [22]. Calculation of the Y-coordinate (CH-coordinate) was performed automatically. The CH coordinate values were calculated as a distance between the CH values calculated using PONDR^®^ online service (http://www.pondr.com/) and the linear border between IDPs and structured proteins (*y* = 2.743 × *x* − 1.109) [41]. Values of the X-coordinate (CDF) were the average of the vertical distances from the CDF curve to the seven boundary points. To obtain CDF-values, the version VSL2 PONDR was used [42].

### 3.4. Prediction of Secondary Structure

Jpred4 (http://www.compbio.dundee.ac.uk/jpred/) was used for prediction of secondary structure for each sequence in our dataset [43].

### 3.5. Analysis and Visualization

Algorithms of search, collection, representation and analysis by the described methods of the data were realized using the freely available programming language Python 3 (https://www.python.org/). The result of the obtained two-dimensional array of data (for CH-CDF plots) was visualized using the Matplotlib library.

## 4. Conclusions

In this work, we show that S1 proteins belong to a unique family, which differs in the classical sense from proteins with tandem repeats. We found that the one-domain and two-domain containing S1 proteins apparently have more stable and rigid structure. An increase in the number of structural domains contributes to the possible transition of a portion of proteins from the folded state to the MG state. For example, for three- and four-domain containing proteins, the ratio of predicted MG state is about 70%. A relatively small percentage of internal flexibility/disorder within individual structural domains could be seen as an indicator of the stability of the S1 domain as one of the OB-fold in this family. At the same time the ratio of flexibility in the separate domains apparently is related to their roles in the activity and functionality of S1. A more stable, compact and conservative central part in the multi-domain proteins is vital for RNA interaction, while terminals domains are for other functions. At the same time, an equal ratio of regions connecting the secondary structure in separate domains and between structural domains indicates about the same organization of multi-domains containing S1 proteins, as well as position and ratio of the secondary structures within separate domains. Reasons for the lack of intact 3D structure of full-length ribosomal protein S1 is not well-understood Perhaps this is due to the high mobility of domains relative to each other in the multi-domain proteins. Further investigation of the flexibility of the available 3D structures for separate S1 domains and the full length S1 domain from *E. coli* in complex with 70S ribosomal subunit will allow finding an accurate explanation.

## Figures and Tables

**Figure 1 ijms-20-02377-f001:**
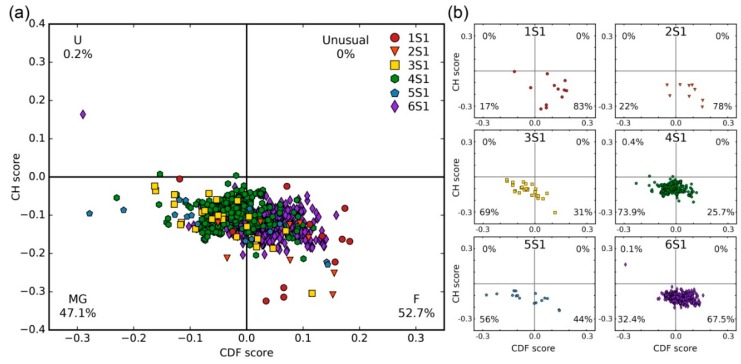
(**a**) Binary disorder analysis (charge–hydrophobicity plot cumulative distribution function (CH-CDF) plots [22]) of 1374 S1 proteins; (**b**) separate S1 proteins groups containing different numbers of structural domains.

**Figure 2 ijms-20-02377-f002:**
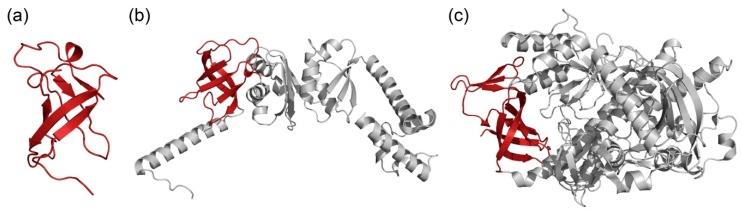
Protein structures with the S1 domain from different bacterial proteins. The S1 domain in each structure is highlighted with red color. (**a**) S1 domain PNPase, PDB code: 1sro; (**b**) antitermination protein NusA, PDB code: 5ml9; (**c**) Ribonuclease R, PDB code: 5xgu.

**Table 1 ijms-20-02377-t001:** Intrinsic flexibility and disorder of S1 protein family and its structural domains. The largest and smallest values are highlighted in bold.

Number of Structural S1 Domains	FoldUnfold (11 aa)			FoldUnfold (5 aa)			IsUnstruct		
	% Disorder for Each Domain		Full Length Proteins	% Disorder for Each Domain		Full Length Proteins	% Disorder for Each Domain		Full Length Proteins
1S1	20 ± 3		25 ± 10	17 ± 11		22 ± 13	17 ± 11		24 ± 17
2S1	1	16 ± 1	20 ± 11	1	13 ± 6	20 ± 5	1	18 ± 5	19 ± 10
	2	24 ± 10		2	20 ± 10		2	28 ± 11	
3S1	1	17 ± 1	15 ± 9	1	20 ± 6	26 ± 7	1	36 ± 13	26 ± 9
	2	21 ± 7		2	21 ± 7		2	36 ± 16	
	3	0		3	13 ± 6		3	20 ± 4	
4S1	1	21 ± 5	18 ± 5	1	25 ± 7	**30 ± 4**	1	24 ± 9	22 ± 5
	2	18 ± 1		2	13 ± 5		2	24 ± 5	
	3	21 ± 6		3	17 ± 8		3	28 ± 10	
	4	18 ± 3		4	16 ± 7		4	23 ± 7	
5S1	1	21 ± 3	17 ± 13	1	22 ± 12	**30 ± 11**	1	28 ± 16	21 ± 15
	2	21 ± 5		2	15 ± 8		2	23 ± 13	
	3	20 ± 3		3	22 ± 12		3	28 ± 13	
	4	24 ± 1		4	22 ± 8		4	35 ± 16	
	5	18 ± 2		5	22 ± 5		5	28 ± 10	
6S1	1	24 ± 9	**13 ± 4**	1	22 ± 8	27 ± 3	1	27 ± 12	16 ± 4
	2	18 ± 3		2	14 ± 8		2	22 ± 7	
	3	18 ± 4		3	**12 ± 6**		3	21 ± 3	
	4	19 ± 3		4	19 ± 6		4	24 ± 4	
	5	20 ± 5		5	27 ± 7		5	25 ± 5	
	6	22 ± 7		6	32 ± 9		6	**45 ± 19**	

**Table 2 ijms-20-02377-t002:** Intrinsic flexibility and disorder of S1 domains in some bacterial proteins.

Protein Name	Source Organism	UniProt Code	Percent of Flexibility/Disorder		
			FoldUnfold (11 aa)	FoldUnfold (5 aa)	IsUnstruct
S1 domain PNPase	*E. coli*	P05055	0	17	17
Protein YhgF	*E. coli*	P46837	0	0	11
Antitermination protein NusA	*E. coli*	P0AFF6	0	36	26
Ribonuclease R	*E. coli*	P21499	13	6	27
Ribonuclease E	*E. coli*	P21513	0	20	26
Tex-like protein N-terminal domain protein	*Kingella denitrificans*	F0F1S0	0	0	13

**Table 3 ijms-20-02377-t003:** Ratio of regions connecting elements of the secondary structure according to the JPred predictions.

Number of Structural S1 Domains	JPred			
	% aa in the Regions Connecting Secondary Structures (Separate Domains)		% aa in the Regions Connecting Secondary Structures (Full Length Proteins)	% aa of the Linkers between Structural Domains (Full Length Proteins)
1S1	41 ± 6		49 ± 7	45 ± 13
2S1	1	50 ± 6	47 ± 3	33 ± 13
	2	41 ± 4		
3S1	1	48 ± 3	46 ± 4	38 ± 7
	2	44 ± 10		
	3	43 ± 3		
4S1	1	47 ± 2	51 ± 2	38 ± 7
	2	47 ± 6		
	3	51 ± 4		
	4	44 ± 2		
5S1	1	49 ± 3	53 ± 4	33 ± 8
	2	51 ± 4		
	3	51 ± 5		
	4	48 ± 3		
	5	48 ± 6		
6S1	1	47 ± 2	52 ± 2	27 ± 3
	2	52 ± 5		
	3	49 ± 3		
	4	50 ± 4		
	5	51 ± 4		
	6	47 ± 3		

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
