# Peer review of "Repeats in S1 Proteins: Flexibility and Tendency for Intrinsic Disorder"

_ijms, 2019, doi:10.3390/ijms20102377_

Round 1
Reviewer 1 Report
The topic of this work is very interesting. The authors have investigated the intrinsic disorder for S1 protein both in single domain and within multiple domains, and can provide some new insights into how polymerization affect the intrinsic disorder, and their possible biological meaning. The method used in this paper is reasonable and credible, but the biological meaning of the result is not so clear. Overall, I suggested to add more discussion for this part.
In detail, normally, the prediction of disorder only depends on sequence. So, for repeat proteins, the percentage of disorder ration should be same for one domain and multiple domains. The results of this work that less disorder in multi-domain proteins may be caused by their structural folds in protein complexes. I was wondering if authors could give a connection between this result with structural aspect, such as OB-fold
“more stable and compact central part in the multi-domain proteins is vital for RNA interaction, terminals domains - for other functions”
this result seems not so new, could you give more explanations of this result?
For ref [6], I was wondering if this paper has been published? if not, a bit more detail information needs to be provided?
Author Response
Reviewer: 1
The topic of this work is very interesting. The authors have investigated the intrinsic disorder for S1 protein both in single domain and within multiple domains, and can provide some new insights into how polymerization affect the intrinsic disorder, and their possible biological meaning. The method used in this paper is reasonable and credible, but the biological meaning of the result is not so clear. Overall, I suggested to add more discussion for this part.
Answer: According to the recommendation of the reviewer we have added more discussion about the biological meaning of the results concerning to some structural aspects of S1 domain as one of the variant of the OB-fold and connection between the ratio of flexibility in the separate domains and full-length proteins and their roles in the activity and functionality of S1 (section 2.4).
In detail, normally, the prediction of disorder only depends on sequence. So, for repeat proteins, the percentage of disorder ration should be same for one domain and multiple domains. The results of this work that less disorder in multi-domain proteins may be caused by their structural folds in protein complexes. I was wondering if authors could give a connection between this result with structural aspect, such as OB-fold.
Answer:
Really, S1 domain is one of the structural versions of the OB-fold, which is considered to be one of the “most ancient” protein folds tolerant to mutations and able to accommodate to the binding of a large number of ligands (Bycroft M, Hubbard TJP, Proctor M, Freund SM V, Murzin AG. The solution structure of the S1 RNA binding domain: A member of an ancient nucleic acid-binding fold. Cell 1997;88(2):235–242). The structure of individual S1 domains (from the ribosomal S1 and various bacterial, archaeal and eukaryotic proteins) is a β-barrel with an additional α-helix between the third and fourth β-sheets; the main function of this domain is RNA-binding. However, at the same time, for full-length, intact ribosomal proteins S1, containing different number of structural domains as well as for eukaryotic proteins (7-15 S1 repeated domains), there are currently no determined structures. Functions of separate S1 domains in these proteins are not defined. This fact may be due to the increased flexibility of multidomain proteins. Our results confirm the fact that the individual S1 domains of the S1 ribosomal protein family are compact and stable, with relatively short loops within the domains. At the same time, the data obtained indicate about the same organization of separate S1 domains and multi-domain S1 proteins. Thus, one of the reasons for the absence of allowed three-dimensional structures of multi-domain proteins may be the mobility of domains relative to each other due to the flexibility of interdomain linkers. Wherein this family apparently having repeats is closed to a “beads-on-a-string” organization with each repeat being folded into a globular domain (similar to the structure of Ig). At the moment, we carry out the investigation of tertiary flexibility predisposition of S1 domains from different proteins for searching of the functional disordered regions potentially involved in interaction natural binding partners.
“more stable and compact central part in the multi-domain proteins is vital for RNA interaction, terminals domains - for other functions” this result seems not so new, could you give more explanations of this result?
Answer:
In fact, only for the well-studied 30S ribosomal protein S1 with six S1-domain repeats from E.coli the biochemical experimental study of various fragments allowed establishing the functions of individual protein domains and parts. For example, it has been shown that cutting one S1-domain from the C-terminus or two S1-domains from the N-terminus of the protein reduces only the effectiveness of protein functions but not its function abilities; the sixth domain is bound with the process of autoregulation of synthesis; cut off the fifth and sixth domain lead to effective participation of remaining part of protein only in synthetic mRNA translation (Amblar M. et al., Rna 2007; Boni I V et al., J Bacteriol 2000).
Our results indicate about the same organization of separate S1 domains and full-length proteins (conservative secondary structure, ratio of linkers and regions connecting secondary structures within domains). In addition, percent of intrinsic flexibility is less for the central domains in the multi-domain proteins. These facts allow us to assume that for all multi-domain S1 proteins more stable and compact domain are located in the central part and is vital for RNA interaction, and more flexible terminals domains - for other functions.
For ref [6], I was wondering if this paper has been published? if not, a bit more detail information needs to be provided?
Answer: At the moment, for reasons beyond the authors' control, unfortunately, the review ref [6] is delayed; if the status of ref [6] is changed, the output data will be communicated to the IJMS editorial staff. According to the recommendation of the reviewer we have added more detail information about the obtained results in ref [6], related to this research (Introduction section).
Reviewer 2 Report
In the present work, Authors studied the tendency for intrinsic disorder and structural characterization of the bacterial S1 proteins within and between structural domains using S1 sequences extracted from UniProt. According to my opinion, later information will be crucial for better understanding of this family of proteins. However, the article is lacking clarity at few places. My minor suggesions are as follows:
Introduction section requires rewritting and rephrasing of sentences, for example -line 33-34 in introduction “ It is known that multi-domain proteins are frequently characterized by occurrence of domain repeats in proteomes across the three domains of life”. Which three domains of life authors are talking about? can you expend it? line 36-39 ”in addition …”. and Line 46-49 “Recently we have shown..."the sentences are very confusing and long.
Result section 2.1: line 88-89 “ In the case of classes S1 proteins containing one, two of six structural domains the ordered state is prevailing (83%, 78% and 67%, respectively).” is it about domain 1, 2 and 6?
CH-CDF analysis shows that six domain shows 67% ordered state and also largest disorder in the IsUnstruct analysis. can you clarify it in results section? Can you please add a short explanation about the difference in values obtained with FoldUnFold (11 aa abd 5 aa) and InUnstruct.
Author Response
Reviewer: 2
In the present work, Authors studied the tendency for intrinsic disorder and structural characterization of the bacterial S1 proteins within and between structural domains using S1 sequences extracted from UniProt. According to my opinion, later information will be crucial for better understanding of this family of proteins. However, the article is lacking clarity at few places.
My minor suggesions are as follows:
Introduction section requires rewritting and rephrasing of sentences, for example -line 33-34 in introduction “ It is known that multi-domain proteins are frequently characterized by occurrence of domain repeats in proteomes across the three domains of life”. Which three domains of life authors are talking about? can you expend it? line 36-39 ”in addition …”. and Line 46-49 “Recently we have shown..."the sentences are very confusing and long.
Answer: We have done.
line 33-34: ” It is known that multi-domain proteins are frequently characterized by occurrence of domain repeats in proteomes across the three domains of life: Bacteria, Archaea and Eukaryotes.”
line 36-38: “In addition, due to their flexibility, domain repeats could be found in cytoskeleton proteins, proteins responsible for transport and cell cycle control”.
line 47-50 “Recently we have shown that the sequence alignments of S1 proteins between separate domains in each group is revealed a rather low percentage of identity. Data obtained indicated that for general functioning of these proteins the structure scaffold (OB-fold) is obviously more important than the amino acid sequence”.
Result section 2.1: line 88-89 “ In the case of classes S1 proteins containing one, two of six structural domains the ordered state is prevailing (83%, 78% and 67%, respectively).” is it about domain 1, 2 and 6?
Answer: We have done.
line 89-90: “In the case of classes S1 proteins containing one, two of six structural domains (1S1, 2S1, 6S1) the ordered state is prevailing (83%, 78% and 67%, respectively)”.
CH-CDF analysis shows that six domain shows 67% ordered state and also largest disorder in the IsUnstruct analysis. can you clarify it in results section? Can you please add a short explanation about the difference in values obtained with FoldUnFold (11 aa abd 5 aa) and InUnstruct.
Answer:
1. Really, the CH-CDF analysis shows that group of six domain containing full length S1 ribosomal proteins is predicted in 67% of cases as the ordered state. Wherein the data obtained from the IsUnstruct and FoldUnfold programs which are given in the Table are in a good agreement with the CH-CDF analysis as far as for the group of six domain containing full length S1 ribosomal proteins the predicted data by the IsUnstruct program showed low percent of disorder (16%). 45% (the largest disorder in the IsUnstruct analysis) is revealed for separate the sixth domains for this group of proteins.
2. The FoldUnfold and IsUnstruct programs are based on the different algorithms mentioned in the section 3.3.1. Short explanation about the difference in values obtained with FoldUnfold (11 aa and 5 aa) was added in the section 3.3.1.
Round 2
Reviewer 1 Report
I sugest that the current version can be accepted